# Position: When AI Decides Who Gets an Organ: Multi-Agentic AI Systems in Transplant Medicine Risk Amplifying Disparities Without Targeted Explainability and Deployment Strategies

**Divya Sharma** [1] [2]  **Ghazal Azarfar** [2]  **Bima J. Hasjim** [3]  **Mamatha Bhat** [2]

## Abstract

Agentic AI systems particularly those built on large language models (LLMs) and deployed as autonomous, role-specialized agents are rapidly emerging in clinical decision-making. This position paper argues that without equity and explainability as core design constraints, such systems will exacerbate healthcare disparities. Using empirical evidence from a multi-agent simulation of a liver transplant selection committee, we demonstrate that even high-performing agents can systematically disadvantage patients based on sex, ethnicity, and socioeconomic status. These disparities arise from agents' reliance on non-clinical proxy variables (insurance type, education level, area deprivation index) and are compounded by the lack of case-level explanations and temporally grounded reasoning. We further contend that without fairness-aware deployment strategies, such systems cannot be reliably audited or ethically integrated into real-world care. In response, we propose a technical roadmap with subgroup-sensitive learning objectives, counterfactual reasoning modules, clinician-in-the-loop governance, and deployment protocols that address the digital divide. We urge the machine learning community to center explainability and health equity in the development and deployment of agentic AI for medicine especially in high-stakes domains where algorithmic decisions may determine who lives and who does not.

[1]Department of Mathematics and Statistics, York University, Toronto, Canada [2]Toronto General Hospital, University Health Network, Toronto, Canada [3]Department of Medicine, University of California, Irvine, USA. Correspondence to: Divya Sharma <divya03@yorku.ca>.

*Proceedings of the 43$^{rd}$ International Conference on Machine Learning*, Seoul, South Korea. PMLR 306, 2026. Copyright 2026 by the author(s).

## 1. Introduction

The emergence of agentic artificial intelligence (AI) where large language models (LLMs) are instantiated as autonomous, role-specialized agents capable of reasoning, collaboration, and decision-making marks a paradigm shift in how machine learning systems interface with high-stakes domains such as medicine (Thirunavukarasu et al., 2023). These agentic systems are increasingly being tested in clinical contexts where complex judgment, temporal reasoning, and multidisciplinary input are essential (Wang et al., 2025; Tariq, 2024; Suura, 2025; Rahsepar Meadi et al., 2025). One critical application domain is transplant selection, where multidisciplinary committees assess transplant eligibility including decisions to remove patients from the waitlist often carrying life-or-death consequences (Tanaka et al., 2025). However, the transplant system has long been critiqued for systemic inequities, disproportionately disadvantaging Black patients and women in access to waitlisting, time-to-transplant, and post-transplant outcomes (Mathur et al., 2010; 2011; Glorioso, 2021).

Recent advances have demonstrated that LLM-based agents can simulate multiple roles, generate structured clinical reasoning, and even reach consensus through structured deliberation (Chen et al., 2025; F Paulo et al., 2025). This holds promise for improving scalability, standardization, and access to expert-level insights in resource-constrained environments. However, without targeted strategies for equitable deployment and explainability, agentic AI systems risk amplifying the very disparities they aim to mitigate. Our development and analysis of one such multi-agent system within liver transplant scenario revealed systematic disparities in decision-making for subgroups including women, Hispanic patients, and individuals from lower socioeconomic backgrounds. Disparate Impact (DI) scores fell below accepted thresholds, and feature attribution analyses revealed reliance on social proxies like education level, insurance status, and Area Deprivation Index (ADI) variables that correlate with race and class, rather than clinical need. Moreover, the decentralized structure of agentic systems where each agent operates autonomously within its domain risks creating fragmented and opaque pipelines, with little visibility into how

individual biases aggregate at the system level.

More critically, these systems do not merely reproduce disparities they risk codifying unspoken institutional value systems into clinical AI. In real-world transplant allocation, for instance, pediatric patients are explicitly prioritized, reflecting a normative stance on age-based vulnerability. Yet there are no parallel mechanisms to account for the caregiving responsibilities of adult women such as mothers of young children (Garg et al., 2000) or the structural disadvantages faced by Black patients who may encounter delayed referral or evaluation (Ifudu et al., 1999). These gaps reflect broader assumptions about whose lives and roles are deemed more "salvageable" or socially valuable. If AI agents learn from historical decisions without interrogating the values embedded within them, they risk encoding biased hierarchies of worth into automated judgment amplifying the invisibility and disadvantage already faced by underserved groups.

At the same time, the digital infrastructure required to deploy agentic AI real-time EHR access, clinician oversight, institutional readiness is unevenly distributed across health systems. Without targeted deployment strategies, the benefits of AI will accrue to well-resourced centers, while under-resourced settings fall further behind.

**This paper takes a position: unless fairness and explainability are treated as core design constraints in the design and deployment of agentic AI, especially in high-stakes domains like transplant medicine, these systems will exacerbate healthcare disparities and encode institutional biases under the guise of objectivity.** We substantiate this position with empirical findings from our multi-agent study simulating liver transplant committee decision-making, alongside an analysis of critical failure modes in data modeling, agent design, and deployment infrastructure.

Our contributions are threefold: (1) We identify mechanisms of bias inherent in multi-agent architectures operating on historical data, (2) We argue for explainability-by-design, especially in systems that simulate human clinical judgment, (3) We propose a technical roadmap for targeted deployment that centers fairness, auditability, and institutional equity.

Ultimately, as agentic AI systems take on greater responsibility in medical decision-making, they will not just reflect clinical knowledge, they will shape it. Ensuring they do so equitably is a challenge the machine learning community must urgently confront.

## 2. Background: Promise and Pitfalls of Agentic AI in Medicine

The rise of agentic AI systems in which large language models (LLMs) are instantiated as autonomous, role-specialized agents has introduced a powerful new paradigm in machine learning (Wang et al., 2025; Tariq, 2024; Suura, 2025; Rah-

separ Meadi et al., 2025; Chen et al., 2025; F Paulo et al., 2025; Liu et al., 2023). These agents go beyond static predictions: they reason over temporally evolving data, simulate domain expertise, coordinate with other agents or users, and navigate complex decision processes. This emergent capability is particularly compelling in medicine, where clinical reasoning is often distributed across multidisciplinary teams and unfolds over time. Technical advances in multi-agent orchestration (e.g., CrewAI, LangChain, AutoGen) (CrewAI contributors, 2024; Auffarth, 2023; Wu et al., 2023) have made it possible to create AI agents that mirror roles like clinicians, nurses, or social workers, interacting with each other through structured dialogues and memory-driven prompts. These frameworks promise to automate workflows, and reduce variation in clinical decision-making particularly in high-stakes osettings. Yet despite these technical strides, the integration of agentic AI into healthcare systems remains fraught with fundamental concerns.

### 2.1. Explainability and Trustworthiness

Agentic systems often function as black boxes. While individual predictions may appear reasonable, the rationale behind agent decisions especially across sequential or collaborative steps remains opaque. Standard tools such as feature attribution or SHAP values are poorly suited for the path-dependent, role-differentiated logic that agentic AI entails. As Alzetta et al. (Alzetta et al., 2020) argue, explainability in multi-agent systems must be grounded in temporal coherence, inter-agent dependencies, and user-aligned rationales. In clinical contexts, this lack of transparency undermines trust, impedes oversight, and creates risks in auditability.

### 2.2. Fairness and Structural Bias

Healthcare datasets reflect decades of systemic inequality. Variables such as insurance status, zip code, and care utilization are deeply correlated with race, gender, and socioeconomic status (Schulman et al., 1995). When agentic AI learns from such data especially when agents are assigned socially sensitive domains like "risk assessment" or "social support evaluation" these biases can become baked into role-specific policies. Research has shown that even widely deployed models underperform or misallocate care for historically marginalized populations (Obermeyer et al., 2019; Seyyed-Kalantari et al., 2020; Grote & Keeling, 2022). In agentic settings, where no single agent sees the full context, these biases can propagate silently across the decision chain.

### 2.3. Deployment Divide and Health System Readiness

Agentic AI systems typically require access to high-quality, structured clinical data, integration with electronic health records (EHRs), sufficient compute infrastructure, and trained clinical champions to oversee operation. These pre-

requisites are often concentrated in tertiary or academic health centers. In contrast, rural hospitals, safety-net systems, and under-resourced clinics face infrastructural and workforce barriers to AI adoption. As highlighted by Rajpurkar et al. (Rajpurkar & Topol, 2025), without proactive strategies for equitable deployment, the diffusion of agentic AI risks entrenching a digital divide where advanced AI augments care only for those already advantaged.

## 3. How Agentic AI Performs in Practice: A Case Study in Transplant Eligibility

This paper argues that current approaches to building and deploying agentic AI in clinical settings are insufficiently robust to prevent bias propagation, opaque decision logic, and inequitable access. We support this position using evidence from a multi-agent simulation conducted in the context of liver transplant eligibility assessment. In our study, we developed a multi-agent system composed of four domain-specialized LLM agents, each emulating a distinct clinical role commonly represented on transplant selection committees: transplant hepatologist, transplant surgeon, cardiologist, and social worker. After evaluating four LLM baselines (Copilot, Gemini, Grok, and GPT-4o) for this task (Appendix Table 1), we instantiated the committee agents using OpenAI's GPT-4o architecture. The temperature setting for the clinical agents was 0.7, reflecting default OpenAI tuning for agent behavior, while a separate medical scribe agent responsible for generating structured patient vignettes from registry data used a temperature of 0.1 to minimize hallucinations and ensure factual accuracy. Prompts were instruction-tuned for each role using guideline-derived templates and expert-authored examples. The agents ingested natural language vignettes serialized from structured data in the Scientific Registry of Transplant Recipients (SRTR), including demographics, MELD scores, comorbidities, and social determinants of health (insurance, education, Area Deprivation Index).

Each agent made independent transplant eligibility decisions, with the hepatologist agent serving as a tie-breaker in cases of discordance, mimicking real-world committee dynamics. Final committee-level decisions were reached via majority voting. Experimental implementation used Python-based orchestration frameworks (CrewAI v0.63.6, LangChain, and AgentOps), and evaluations spanned 8,412 liver transplant cases from the SRTR dataset ($N = 7,033$ eligible/waitlisted; $N = 1,379$ with assigned contraindications). Synthetic contraindication profiles were programmatically assigned to a subset of cases to simulate ineligible scenarios. To note, our study uses a hybrid empirical-simulation design: real national registry patients and outcomes, with synthetic negative-case construction only where registry structure makes declined candidates unavailable. We evaluated alternative prompting strategies (zero-shot, zero-shot chain-of-thought, and self-consistency) prior to selecting our final configuration zero-shot-CoT and self-consistency (Appendix Table 2). The primary outcomes assessed were: (1) accuracy in identifying absolute contraindications to transplant, and (2) accuracy in predicting 6- and 12-month post-transplant survival. These outcomes were chosen to reflect the operational criteria used by human transplant selection committees. Also, The AI-SC is presented here as a decision-support stress-test and risk-discovery framework, not as a replacement for human transplant committee judgment. While overall performance was high (98.2% for contraindication detection and $> 94\%$ for predicting 6- and 12-month survival), our fairness audits uncovered systematic disparities detailed below.

### 3.1. Failure Mode 1: Role-Specific Bias Propagation

Despite high overall accuracy, our multi-agent AI system demonstrated systematic disparities in decision-making across demographic and socioeconomic subgroups. This phenomenon was particularly evident in role-specialized agents, where domain-specific reliance on proxy variables amplified structural biases embedded in the training data.

In our simulation of a multidisciplinary liver transplant selection committee, each agent representing a clinician (hepatologist, surgeon, cardiologist) or a social worker exhibited distinct patterns of feature dependence. The "social worker" agent, for example, disproportionately weighted persistent non-compliance, insurance status, education level, and the Area Deprivation Index (ADI) (Figure 1(f)). These variables, while clinically relevant in some contexts, are known proxies for socioeconomic disadvantage and race.

To assess equity in eligibility classification, we conducted a Disparate Impact (DI) analysis (Briscoe & Gebremedhin, 2024) across protected attributes including sex, race, ethnicity, education, liver disease etiology, and ADI-based socioeconomic strata (Figure 1(a-d)). DI was computed as the ratio of favorable classification rates for a given subgroup to a privileged reference group (e.g., female vs. male, hispanic vs. non-hispanic).

Several subgroups exhibited DI scores well below the standard fairness threshold (DI ¡ 0.80), including female patients, Black individuals, patients with high school education, and those from the most socioeconomically deprived quintiles. This suggests that role-specific feature reliance led to lower eligibility recommendations for marginalized populations.

The radar plot in Figure 1(e) illustrates intersectional disparities across multiple attributes, underscoring the need for subgroup-specific auditing pipelines and fairness constraints during model development. These findings illustrate a critical failure mode of agentic AI: when trained on historical

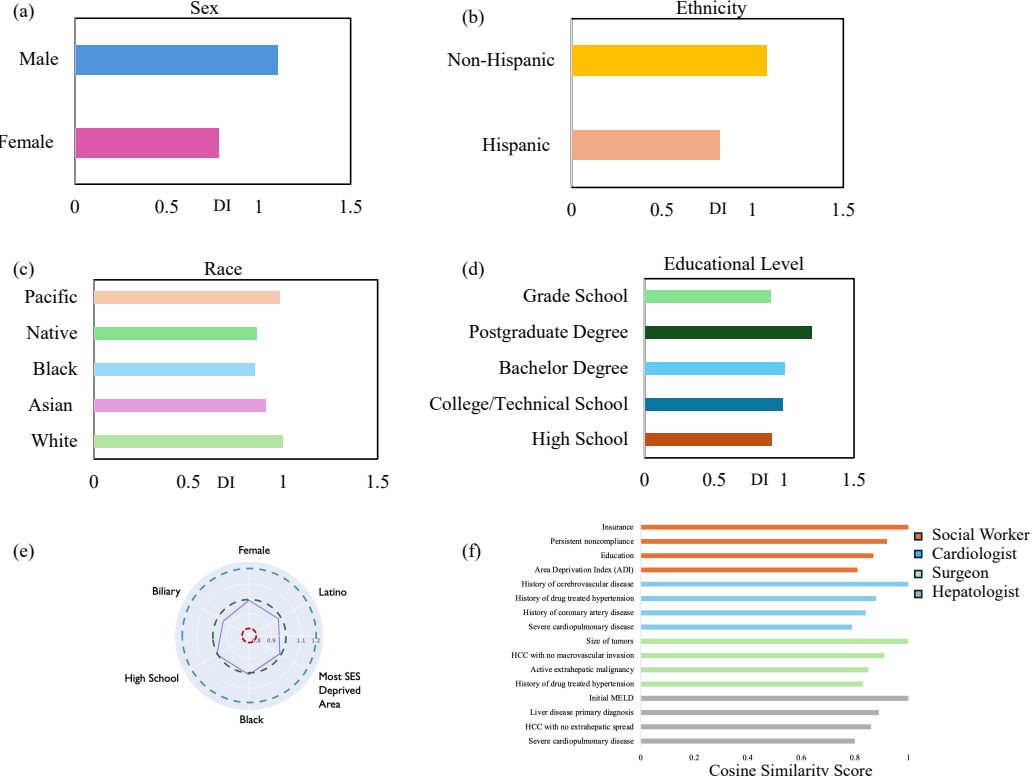

*Figure 1.* (a–d) Disparate Impact (DI) scores by sex, ethnicity, race, and education show consistent under-classification of disadvantaged groups. (e) Radar plot highlights intersectional disparities across key subgroups. (f) Cosine similarity scores indicate agent-specific reliance on clinical and social features in decision-making.

data without explicit fairness constraints, role-specialized agents may institutionalize unjust patterns of exclusion.

## 3.2. Failure Mode 2: Opaque Decision Logic Undermines Auditing and Equity

In high-stakes clinical contexts, explainability is not a luxury, it is a prerequisite for trust, safety, and regulatory viability. Yet most agentic AI systems lack mechanisms for case-level transparency when operating as ensembles of agents with distinct roles and interdependent logic.

This limitation was evident in our liver transplant simulation. While population-level analyses (cosine similarity between decision vectors and input features) revealed which variables were most influential for each agent, the "cardiologist" emphasizing cardiovascular history, the "surgeon" focusing on tumor size, these methods failed to generate actionable, case-specific justifications (Figure 1(f)). For instance, when the "social worker" agent rejected a borderline candidate, the system offered no counterfactual insight: which feature drove the decision? Would modifying a non-clinical attribute (e.g., education level) have changed the outcome?

This lack of transparency critically impedes fairness audit-

ing. Without traceable decision paths or post hoc explanations, it is impossible to determine whether an agent's output was justified, biased, or erroneous, particularly when bias arises from proxy variables embedded in social determinants. The challenge grows in multi-agent systems, where one agent's decision may depend implicitly on the outputs or assumptions of others, creating hidden chains of influence defying single-agent attribution.

These silent failure modes are not benign rather disproportionately affect already marginalized populations such as women, racialized groups, and patients from lower socioeconomic strata who are less likely to appeal adverse outcomes and more likely to be harmed by algorithmic opacity.

Critically, the absence of explainability also places such systems at odds with emerging regulatory standards. Frameworks like the EU AI Act and the U.S. FDA's proposed guidelines for AI in medical devices increasingly require that automated clinical systems produce interpretable, auditable rationales for their decisions. Agentic AI systems that cannot meet these standards risk exclusion from clinical workflows, not because of poor accuracy, but due to ethical and legal misalignment.

To be clinically viable and socially responsible, multi-agent systems must include built-in capabilities for trajectory-aware explanations, feature-level attribution, and interactive counterfactual reasoning. Without such mechanisms, bias becomes uncheckable and accountability becomes diffuse.

### 3.3. Failure Mode 3: The Digital Divide and Naïve Deployment

Even if agentic AI systems are fair in design and interpretable in function, their impact hinges on equitable deployment, an area where current implementation strategies fall short. In our experiment, performance depended on infrastructure that is unevenly distributed across health systems: structured electronic health record (EHR) pipelines, fine-tuned large language models (LLMs), cloud compute environments, and clinician-users familiar with AI tools.

This asymmetry poses a structural risk: without proactive deployment planning, agentic AI will reinforce existing institutional hierarchies offering enhanced decision support to well-resourced academic centers while leaving community hospitals and safety-net providers behind. These settings often lack the technical capacity or capital to deploy, calibrate, and maintain complex multi-agent systems.

Naïve, one-size-fits-all deployment further compounds the problem. Agentic models trained on data from specific geographies or patient cohorts may perform poorly in settings with different care patterns, documentation norms, or demographic compositions. Without local adaptation such as subgroup reweighting, transfer learning, or calibration pipelines these systems risk amplifying misclassifications in underrepresented populations.

Cost barriers add another layer of exclusion. Proprietary LLM agents require significant investment in API usage, licensing, model updates, and operational oversight. For underfunded institutions, these expenses are often prohibitive. In the absence of open-source implementations or subsidized deployment strategies, the gap between digitally mature and resource-constrained health systems will widen further entrenching inequities in AI-supported care.

Ultimately, deployment cannot be an afterthought. Equity must be engineered not just into the model architecture but into the deployment protocols, funding structures, and regulatory frameworks that govern AI diffusion in medicine.

## 4. A Roadmap for Targeted Deployment and Explainable Multi-Agent AI Design in Medicine

To responsibly deploy multi-agent AI systems, we outline a technical roadmap with five core pillars integrating explainability, fairness, and deployment feasibility as foundational design constraints rather than retrospective modifications (Figure 2). The proposed architecture is a staged menu, not a single deployment requirement: a minimum viable fairness tier, subgroup auditing, disparate-impact monitoring, clinician override, and audit logging, is feasible for lower-resource centers, whereas components such as Bayesian uncertainty heads or structural causal modules represent advanced layers for well-resourced institutions.

### 4.1. Data Modeling: Structured, Fair, and Transparent Inputs

Multi-agent AI systems for clinical decision-making must be built on data pipelines that explicitly account for structural biases and support high-fidelity temporal modeling. We outline four design requirements:

- **Longitudinal feature engineering:** Agentic models must capture temporal evolution in patient states to support reasoning over multi-step clinical trajectories. This includes constructing sliding-window statistics (e.g., exponentially weighted moving averages of lab values (Sukparungsee et al., 2020)), encoding diagnosis or treatment transitions, and integrating time-aware embedding layers. Without such representations, agents risk overfitting to static snapshots of care failing to reflect disease progression or treatment response.

- **Missingness modeling as signal:** Rather than imputing missing values as noise, models should incorporate learnable representations of missingness via GRU-D or attention-based masking (Che et al., 2018; Du et al., 2023). In many EHRs, missingness reflects sociotechnical processes, patients with lower Socioeconomic Status (SES) may have fewer lab tests thus encoding latent disparities.

- **Social determinant normalization:** Variables that encode social context such as Area Deprivation Index (ADI), insurance status, and education level must be explicitly labeled during preprocessing as sensitive or bias-prone features. This "flagging" enables downstream models to treat these inputs differently: for instance, by minimizing their influence via adversarial debiasing (Zhang et al., 2018) or applying fairness-aware regularization in latent space (Yang et al., 2023). Without such mechanisms, these variables may act as proxies for protected attributes (e.g., race, income), resulting in systematic disadvantages during role-specific agent reasoning (e.g., decisions by a "social worker" agent).

- **Data provenance and context tagging:** Structured metadata must track source hospital, documentation norms, and provider type. Embedding this context

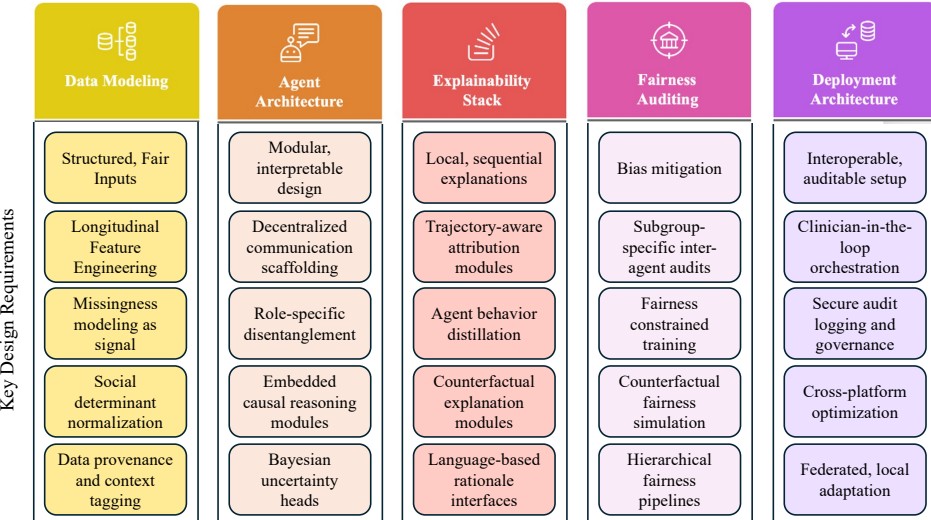

*Figure 2.* A modular roadmap for building fair, explainable, and deployable multi-agent AI systems in medicine. Each column represents a key design dimension, with concrete technical requirements supporting trustworthiness across data, agent reasoning, explanation, bias mitigation, and deployment architecture.

into the data schema via tokenized meta-features or context-aware attention layers (Yang et al., 2019) enables institution-level domain adaptation and supports explainability audits.

These requirements can be operationalized via modular data pipelines using tools such as Apache Beam, TensorFlow Extended (TFX), or PyTorch DataPipes, and evaluated using fairness diagnostics (e.g., disparate impact, equal opportunity) across data slices.

### 4.2. Agent Architecture: Modular, Interpretable, and Role-Aligned

Multi-agent systems must mirror the distributed, asynchronous, and role-specific nature of real clinical committees. The architecture must enable clear decision boundaries, modular interpretability, and inter-agent auditability.

- **Decentralized communication scaffolding:** Implement multi-agent orchestration using frameworks such as LangChain or CrewAI (CrewAI contributors, 2024; Auffarth, 2023), where each agent operates with a distinct observation set and a role-specific action space mirroring real-world clinical specialization. Inter-agent communication should be mediated through structured message-passing protocols (e.g., memory tokens, task-specific APIs (Kim et al., 2021)), designed to emulate clinical hierarchies and consensus workflows, such as escalation to a lead clinician or committee voting.

- **Role-specific disentanglement:** To preserve functional boundaries between agents and prevent deci-

sion homogenization, enforce disentangled representations during training via orthogonality constraints, role-specific latent spaces, or adversarial separation losses (Wen & Yin, 2013; Tsai & Chien, 2017). This ensures each agent contributes domain-specific reasoning, enhances interpretability, and allows attribution of decisions to distinct clinical roles (e.g., "eligibility was downgraded due to concerns raised by the social worker agent").

- **Embedded causal reasoning modules:** Agents must reason beyond statistical associations to assess the impact of interventions. Integrating structural causal models (SCMs), causal graphs, or differentiable modules such as Causal Transformers (Wu et al., 2024) enables agents to perform counterfactual reasoning (via do-calculus (Pearl, 2012)). This empowers agents to simulate outcomes under hypothetical treatments and attribute clinical effects to specific decisions crucial for ethically grounded medical AI.

- **Bayesian uncertainty heads:** Agents should produce both decisions and calibrated uncertainty estimates to reflect confidence levels. This can be implemented using techniques such as Monte Carlo dropout, variational inference, or deep ensembles (Seoh, 2020; Miok et al., 2019). These uncertainty scores are essential for downstream uses including confidence gating, clinician override mechanisms, ensemble arbitration, and risk-aware decision escalation ensuring safe deployment in high-stakes medical contexts.

## 4.3. Explainability Stack: Local, Sequential, and Counterfactual

To ensure safety and trust, multi-agent AI systems must deliver explanations that are temporally grounded, role-disentangled, and clinically actionable. We recommend the following architecture-level components:

- **Trajectory-aware attribution modules:** Equip agents with time-sensitive explanation mechanisms, such as Integrated Gradients (Sundararajan et al., 2017) or Temporal SHAP (Lundberg et al., 2018), to identify which features and timepoints most influenced their decisions. These attributions should be rendered as temporal heatmaps or sequential explanation timelines, enabling clinicians to trace the agent's reasoning across the patient trajectory and validate clinical plausibility.

- **Agent behavior distillation:** Translate complex agent decision policies into interpretable surrogate models (decision trees, symbolic rules, or sparse linear models) through distillation (Coppens et al., 2019). These surrogates should approximate the original agent behavior within a predefined fidelity threshold ($\epsilon$-fidelity (Yagi & Nomura, 2015)), and be validated using metrics such as balanced accuracy, G-mean, or area under the fidelity curve (AUC-F) (Ribeiro et al., 2016). This enables offline auditing, regulatory transparency, and clinician-facing interpretability without compromising core model utility.

- **Counterfactual rationalization layers:** Integrate conditional generative models such as CausalGANs (Kocaoglu et al., 2017) or Counterfactual Transformers (Melnychuk et al., 2022) into the agent pipeline to support "what-if" clinical queries. These models should generate plausible alternate patient profiles conditioned on specific feature manipulations (e.g., gender, insurance status), and simulate corresponding agent decisions. This enables interrogation of decision boundaries and detection of potential bias, such as "Would this patient have been approved if their gender were male, holding all else constant?"

- **Natural language explanation interfaces:** Attach encoder-decoder models (e.g., GPT-4-turbo fine-tuned on radiology/pathology reports) to generate rationale sequences. These modules must support clinician feedback loops and provide provenance (e.g., cited guidelines, referenced lab thresholds).

## 4.4. Fairness Auditing and Bias Mitigation in Multi-Agent Settings

To ensure equitable deployment, multi-agent clinical AI systems must undergo rigorous, multi-level fairness evaluation and mitigation. In contrast to single-model settings, fairness in agentic systems emerges from complex role-based decision pipelines and their interactions. We recommend the following technical strategies:

- **Subgroup-specific inter-agent audits:** Analyze decision distributions and consensus pathways stratified by protected attributes (e.g., race, gender, education). Quantify divergence in agent outputs using conditional parity metrics and disaggregated disparate impact scores across patient subgroups (Koumeri et al., 2023).

- **Fairness-constrained training objectives:** Integrate fairness constraints (e.g., demographic parity, equalized odds (Awasthi et al., 2020)) directly into each agent's loss function. Use in-processing debiasing methods such as adversarial regularization or constrained optimization with Lagrangian multipliers to enforce subgroup fairness at the model level.

- **Counterfactual fairness simulation:** Conduct systematic counterfactual evaluations by simulating patient records with altered sensitive attributes. Use doubly robust estimators and causal inference frameworks (e.g., inverse propensity reweighting, causal forests (Wang et al., 2024)) to assess treatment disparities and decision variability across hypothetical profiles.

- **Hierarchical fairness pipelines:** Implement fairness audits at multiple abstraction levels: individual agent outputs, pairwise agent alignment, and committee consensus. Track divergence across agents in how fairness violations propagate through the deliberation chain, especially in cases involving override or escalation.

## 4.5. Deployment Architecture: Interoperable, Auditable, and Resource-Aware

This subsection addresses the deployment inequities identified in Section 3.3, including uneven infrastructure, proprietary API-cost barriers, and limited local technical support in under-resourced settings. We advocate the following architectural principles for real-world deployment of Agentic AI systems:

- **Clinician-in-the-loop:** Integrate agent outputs with a clinical oversight layer that enables confidence-weighted voting, real-time uncertainty visualization, and structured override mechanisms. Interfaces should support rationale review, allow traceable dissent (via structured annotations), and prioritize clinician arbitration in ambiguous or high-risk cases.

- **Immutable audit logging and governance:** Implement tamper-proof decision logs using cryptographi-

cally signed or blockchain-based architectures to capture agent recommendations, underlying rationales, clinician feedback, and overrides (Thurzo, 2025). These logs serve as a foundation for regulatory compliance (e.g., EU AI Act, FDA SaMD), institutional accountability, and continuous monitoring of agent behavior.

- **Cross-platform optimization:** Apply model compression techniques (such as ONNX export, TensorRT acceleration, quantization-aware training (Zhou et al., 2023)) to enable efficient inference on both edge devices and cloud-based systems. Ensure adaptive execution modes that dynamically switch between local and cloud resources based on latency, compute availability, and security requirements.

- **Federated adaptation and local fine-tuning:** Employ federated learning, differential privacy, and split learning protocols to enable decentralized model updates without centralizing Protected Health Information (PHI) (Ali et al., 2022). Tailor agent behavior to local clinical norms and EHR schemas through site-specific fine-tuning, preserving privacy while improving generalizability and performance across heterogeneous care settings. A minimum viable fairness tier, centered on subgroup auditing, disparate-impact monitoring, and clinician override mechanisms, is feasible for lower-resource centers; more resource-intensive components are intended for well-resourced institutions.

## 5. Alternative Views

A natural alternative perspective is a precautionary one: agentic AI should not be used in transplant selection (or similarly irreversible allocation decisions) until prospective evidence demonstrates benefit and safety. This view is compelling given the stakes and the limits of simulation-based validation. Our position is not that simulation can substitute for clinical evaluation, but that it is a necessary pre-deployment instrument for risk discovery: it can surface subgroup-specific failure modes, proxy reliance, and emergent multi-agent dynamics early before any patient-facing use. In that sense, the disagreement is less about whether prospective evaluation is required (we agree it is), and more about whether equity and transparency should be treated as optional "later-stage" features or as design-time gates that determine what is safe enough to test in the first place.

A second counter-position holds that standardization itself is a fairness intervention: human committees are noisy, inconsistent, and subject to implicit bias, so even an imperfect agentic system may improve equity by reducing inter-clinician and inter-center variability and by making multidisciplinary reasoning available in resource-constrained settings. We agree that reducing unwarranted variation can improve quality and may mitigate certain inequities. However, standardization is not synonymous with equity; it can also reliably reproduce the same structural disadvantage everywhere if the system encodes biased proxies or systematically discounts particular clinical narratives. In allocation settings, scalable errors are often more harmful than idiosyncratic ones, which is precisely why standardized systems require explicit subgroup audits, intersectional evaluation, and contestable decision rationales rather than opacity.

A third view prioritizes performance and feasibility: strict explainability-by-design and fairness constraints may impose an "accuracy tax," add latency, and create workflow friction that undermines adoption. This critique is important and motivates a tiered approach rather than maximal interpretability everywhere. Our claim is not that every case must trigger heavy counterfactual reasoning, but that high-stakes or contested decisions require structured transparency and logging sufficient for clinical review and governance. Similarly, fairness mechanisms need not be implemented only as optimization constraints; they can operate as deployment gates and monitoring practices (subgroup drift checks, targeted calibration audits, periodic review) that preserve clinical utility while preventing silent harm.

Finally, critics note that fairness goals can conflict in scarcity settings: parity metrics may clash with utility objectives, and subgroup constraints can backfire across intersections. We view this not as a reason to avoid fairness, but as a reason to make value choices explicit, test multiple fairness notions, and audit intersectional outcomes over time. Related sociotechnical concerns argue that clinician-in-the-loop oversight can be performative due to cognitive load, alert fatigue, and automation bias. We take this as a design requirement: oversight must be actionable (clear override pathways, concise decision summaries, escalation triggers) and institutional (periodic audits and multidisciplinary governance), rather than relying on clinicians to interrogate dense explanations at the point of care.

## 6. Call to Action

While our roadmap outlines concrete design and deployment strategies to mitigate risks, we close with a broader call to action. This position paper has surfaced three core vulnerabilities in current design and deployment practices: These include biased proxies in historical data, limited trajectory-level explainability, and inequitable deployment that amplifies institutional disparities.

We call on the ML community to engage with the following technical frontiers: (1) Causal and sequential explainability for temporally grounded decisions, (2) Subgroup-aware fairness mechanisms integrated into agent communication

and consensus protocols, (3) Governance frameworks that enable real-time arbitration, override, and escalation in clinical settings and (4) Deployment infrastructure supporting auditability, interoperability, and equitable access.

We also issue role-specific calls to action: (1) To ML researchers: Expand evaluation criteria beyond aggregate accuracy to include interpretability, calibration across subgroups, and real-world resilience (Obermeyer et al., 2019; Seyyed-Kalantari et al., 2020; Briscoe & Gebremedhin, 2024), (2) To industry developers: Prioritize human-in-the-loop interfaces and localized adaptation before broad deployment (Rajpurkar & Topol, 2025) and (3) To policymakers: Enforce explainability-by-design and stratified performance reporting to safeguard against algorithmic harm (Koumeri et al., 2023; Briscoe & Gebremedhin, 2024; Thurzo, 2025).

Agentic AI is not neutral infrastructure it encodes and executes values. If we fail to embed equity and transparency into these systems from the outset, we risk institutionalizing new hierarchies of access, trust, and survival.

## Acknowledgments

We acknowledge the support of the Natural Sciences and Engineering Research Council of Canada (NSERC), grant number RGPIN-2024-05548 for funding this work.

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

# A. Supplementary Appendix

## A.1. Dataset Details

**Data source.** We used the **Scientific Registry of Transplant Recipients (SRTR)**, a national registry maintained under the U.S. Organ Procurement and Transplantation Network (OPTN) that records all donors, waitlisted candidates, and transplant recipients in the United States.

**Study design and study period.** This was a hybrid cohort study of adult ($\geq$ 18 years) liver transplant candidates/recipients spanning **01/01/2004–06/01/2024**. The analytic cohort consisted of **deceased donor liver transplant (DDLT)** recipients with adequate post-transplant follow-up, defined as either death within 1 year post-transplant or at least 1 year of follow-up among survivors.

**Inclusion and exclusion criteria.** We included adult DDLT recipients and excluded patients with previous transplants, living donor liver transplantation, and pediatric recipients, as the organ allocation processes differ from adult DDLT (Figure S1).

**Synthetic contraindications cohort.** Because SRTR primarily includes individuals already waitlisted, to evaluate whether the AI selection committee (AI-SC) could identify candidates deemed ineligible for waitlisting, we generated a hypothetical cohort by randomly assigning known **absolute contraindications to liver transplantation** to a subset of SRTR patients ($N = 1,379$; 16.4%). Assigned contraindications included: metastatic hepatocellular carcinoma (HCC), active extrahepatic malignancy, cardiopulmonary disease, currently septic, active alcohol/drug use, acquired immunodeficiency syndrome (AIDS), persistent noncompliance, and lack of social support. Assignments were not mutually exclusive (patients could receive $> 1$ contraindication). The distribution of extrahepatic malignancy was based on the World Health Organization's most prevalent cancer-related causes of mortality.

**Variables.** A total of **59** equally weighted variables were used to construct clinical vignettes and drive AI-SC decisions, spanning clinicodemographic characteristics, end-stage liver disease (ESLD) features, and social determinants of health (SDOH). SDOH was represented using the **Area Deprivation Index (ADI)**, merged with SRTR at the ZIP-code level (Supplemental Methods).

**Cohort size and geographic coverage.** The final analytic cohort included **8,412** patients: 83.6% waitlisted/eligible ($N = 7,033$) and 16.4% with assigned contraindications ($N = 1,379$).

**Outcome definitions.** The AI-SC was evaluated on three tasks: (i) identifying absolute contraindications, (ii) predicting **6-month** post-transplant survival benefit, and (iii) predicting **1-year** post-transplant survival benefit. For survival endpoints, observed SRTR outcomes were treated as ground truth; for contraindication detection, assigned contraindications defined ground truth.

**Data quality review.** A random sample of **500** generated vignettes was manually reviewed (BJH, FGL), and hallucinations were identified in $< 1\%$ of cases.

## A.2. Supplemental Methods

**Handling missing values.** Redundant variables (e.g., history of diabetes vs. insulin dependence) and variables with $> 20\%$ missingness were excluded to minimize hallucinations. The AI-SC was instructed not to include or make assumptions when encountering missing data during decision-making.

**Merging the Area Deprivation Index (ADI).** ADI is a 17-item composite score from 0–100 that aggregates SDOH variables and provides a standardized ranking to identify disadvantaged census tracts in the U.S. Population-weighted mean ADI values were computed at the 5-digit ZIP code level and merged with SRTR records. Higher ADI indicates lower socioeconomic status (SES). ADI quintiles were categorized as: Least Deprived (0–20), Less Deprived (21–40), Moderately Deprived (41–60), More Deprived (61–80), and Most Deprived (81–100).

**Prompting techniques.** We evaluated three prompting strategies: (i) zero-shot, (ii) zero-shot chain-of-thought (zero-shot-CoT), and (iii) zero-shot-CoT with self-consistency (zero-shot-CoT-SC). CoT promotes step-by-step decomposition for transparency. For self-consistency, each case was assessed five times and the final label was selected by majority (at least 3 of 5), consistent with prior evidence that five samples yields strong gains.

**Temperature settings.** The medical-scribe agent (clinical vignette generation) used temperature $0.1$ to reduce hallucinations. Committee agents used temperature $0.7$, consistent with default agent settings.

**Cosine similarity index (variable-use attribution).** To assess whether each agent emphasized role-relevant inputs, reports were segmented into sentences/sub-sentences and embedded into a vector space. Each SRTR variable was represented by a sentence label (e.g., "Age" $\rightarrow$ "How old is the patient?") and embedded in the same space. For each report segment, cosine similarity was computed against all SRTR labels; the label with the maximum absolute similarity was matched to the segment. Per-variable similarity scores were averaged across cases to produce an overall relevance score for each agent.

**Fairness: demographic parity and disparate impact (DI).** We examined demographic parity as an approximation of fairness: equal probability of assignment to the positive class across groups. Disparate impact was computed as:

$$\mathrm{DI} = \frac{\mathbb{P}(\hat{Y} = 1 \mid G = \text{subgroup})}{\mathbb{P}(\hat{Y} = 1 \mid G = \text{others})} = \frac{TP_{\text{subgroup}} + FP_{\text{subgroup}}}{N_{\text{subgroup}}} \Bigg/ \frac{TP_{\text{others}} + FP_{\text{others}}}{N_{\text{others}}}.$$

DI$> 1$ indicates advantage; DI$< 1$ indicates disadvantage. A common threshold for potential unfairness is DI$\geq 1.2$ or DI$\leq 0.8$.

### A.3. Supplementary Tables

*Table 1.* **Single-LLM evaluation.** Comparative evaluation of candidate LLMs and our proposed multi-agent AI-Selection Committee (AI-SC) for transplant candidacy classification on a random subset of 250 patients. Each system received the same structured clinical vignette format and was required to output a binary accept/reject decision with a brief justification. Performance is summarized using standard classification metrics (accuracy, sensitivity, specificity, precision, recall, F1-score).

| LLM | Accuracy (%) | Sensitivity | Specificity | Precision | Recall | F1-score |
|---|---|---|---|---|---|---|
| Copilot | 58.06 | 0.50 | 1.00 | 1.00 | 0.50 | 0.66 |
| Gemini | 96.77 | 1.00 | 0.80 | 0.96 | 1.00 | 0.98 |
| Grok | 90.32 | 0.88 | 1.00 | 1.00 | 0.88 | 0.94 |
| GPT-4o | 93.54 | 0.92 | 1.00 | 1.00 | 0.92 | 0.96 |
| AI-SC | 98.97 | 0.99 | 1.00 | 1.00 | 0.99 | 0.99 |

*Table 2.* **Performance across prompting methods** on a random subset of 250 patients. CoT = chain-of-thought; SC = self-consistency.

| Method | Accuracy | | | Sensitivity | | | Specificity | | |
|---|---|---|---|---|---|---|---|---|---|
| | Listing | 6 months | 1 year | Listing | 6 months | 1 year | Listing | 6 months | 1 year |
| Zero-shot | 93.6% | 90.8% | 85.2% | 0.99 | 0.99 | 0.99 | 0.82 | 0.65 | 0.53 |
| Zero-shot-CoT | 96.0% | 90.8% | 85.2% | 0.99 | 0.99 | 0.99 | 0.84 | 0.66 | 0.55 |
| Zero-shot-CoT-SC | 97.2% | 92.0% | 86.6% | 0.99 | 0.99 | 0.99 | 0.88 | 0.70 | 0.57 |

