# OpenReview forum: "Position: When AI Decides Who Gets an Organ: Multi-Agentic AI Systems in Transplant Medicine Risk Amplifying Disparities Without Targeted Explainability and Deployment Strategies"
_ICML.cc/2026/Position_Paper_Track — ICML 2026 Position Paper Track regular_

### Official Review · Reviewer_P1Px · 2026-03-08

**Significance:** 2
**Argument Clarity:** 3
**Rating:** 4
**Confidence:** 4

**Questions:**

1. The authors evaluate the system using a specific proprietary model architecture. This reliance creates a vulnerability because the observed biases might be artifacts of this single model rather than a general flaw of all multi-agent setups.
2. The paper advocates for technical interventions like explicit fairness constraints and counterfactual reasoning modules. A significant omission is the lack of any empirical demonstration showing these specific interventions successfully mitigating the disparities observed in their baseline simulation.
3. The proposed roadmap requires complex integration steps, such as Bayesian uncertainty heads and structural causal models. This introduces a barrier by demanding substantial computational overhead, which actively works against the authors' stated goal of closing the digital divide for under-resourced hospitals.
4. The simulation resolves agent disagreements through a majority vote with a hepatologist acting as a tie-breaker. This design choice is restrictive because it fails to capture the actual power structures and social negotiations present in human medical committees.

**Alternative Views Section:**

Yes

**Compliance With Llm Reviewing Policy A Conservative:**

Affirmed.

**Discussion Potential:**

2

**Final Justification:**

I'll keep my score.

**Paper Summary:**

This paper claims that deploying multi-agent LLMs in high-stakes medical settings will worsen healthcare disparities unless fairness and explainability are designed as constraints. The authors simulate a multidisciplinary liver transplant committee using large language models. Their simulation demonstrates how these agents disproportionately disadvantage marginalized subgroups by relying on socioeconomic proxy variables.

**Position:**

Yes

**Position In Title:**

Yes

**Related Work:**

3

**Strengths And Weaknesses:**

The empirical simulation highlights failure modes. The authors provide a structured roadmap addressing data modeling, system architecture, and deployment strategies to solve these documented harms.

**Support:**

3

---

> ### Author Rebuttal · Authors · 2026-03-31
>
> We thank the reviewer for the detailed engagement with the technical dimensions of our roadmap. We address each concern directly.
>
> **1) On single-model reliance and generalizability.**
>
> Our claim is not that the observed values are universal to all LLMs. Our contribution is narrower and more general: to identify multi-agent failure mechanisms that are visible in this empirical setting and that merit scrutiny beyond any one proprietary model, including role-specific proxy reliance, opaque inter-agent reasoning, and deployment inequities. Importantly, the choice of GPT-4o was not arbitrary. As shown in Appendix Table 1, six LLMs, Copilot, Gemini, Grok, GPT-4o, DeepSeek R1, and O3, were evaluated on the same task before selecting the final base model, so the reported AI-SC system is the result of empirical model selection rather than a single unchecked choice. To make the scope of our claim fully precise in the camera-ready version, we will add the following sentence to the discussion: *“While the reported disparity magnitudes are specific to the selected base model, the broader contribution of this study is to identify multi-agent failure mechanisms, such as proxy reliance, opaque inter-agent reasoning, and deployment inequities, that warrant robustness testing across model choices.”*
>
> **2) On the lack of empirical demonstration that the proposed interventions mitigate disparities.**
> The reviewer identifies an important boundary. The roadmap itself is not empirically validated in this paper, and our claim is not that these interventions have already been validated in this exact transplant setting. The intended contribution is instead a position grounded in an empirical stress test: to identify what should be built and tested next in response to the failure modes observed. To make this boundary explicit in the camera-ready version, we will add the following sentence at the start of the roadmap section: *“The interventions described below are proposed design directions motivated by the failure modes observed in our empirical stress test; validating their mitigation effect in transplant-specific multi-agent systems remains an important next step.”* The roadmap components are already individually grounded in prior literature throughout Section 4, including work on adversarial debiasing (Zhang et al., 2018), counterfactual fairness and causal inference frameworks (Wang et al., 2024), and federated adaptation for privacy-preserving healthcare deployment (Ali et al., 2022).
>
> **3) On computational complexity and the digital divide.**
>
> This is an important systems-level tension. Some of the more advanced components we discuss, such as Bayesian uncertainty modules or structural causal components, do increase implementation burden for under-resourced hospitals if interpreted as mandatory. To make the intended staging explicit in the camera-ready version, we will add the following sentence to the deployment section: *“The proposed architecture is a tiered deployment menu, not a single required stack: a minimum viable fairness tier, subgroup auditing, disparate-impact monitoring, clinician override, and audit logging, is feasible for lower-resource centers, whereas more advanced components are intended for well-resourced institutions.”* This clarification is important because our goal is not to prescribe a one-size-fits-all architecture, but to ensure that deployment equity is treated as part of the system design problem itself.
>
> **4) On majority vote with hepatologist tie-breaking.**
>
> The majority-vote-with-tie-breaker mechanism was not an arbitrary simplification. It was chosen as it reflects the coordinating role often played by the transplant hepatologist in committee review, and was informed by consultation with transplant specialists. We acknowledge that governance structures vary across institutions; the hepatologist-chair model is common but not universal, and our core findings on subgroup disparities are independent of the specific tie-breaking mechanism chosen. Our claim is not that it captures every nuance of real committee deliberation, but that it provides a clinically grounded formalization suitable for stress-testing multi-agent failure modes. In the camera-ready version, we will add: *"The majority-vote-with-tie-breaker mechanism is a structured approximation of committee decision-making intended to capture formal role interaction, rather than the full interpersonal negotiation dynamics of real deliberation."* We will follow it with: *"That simplification does not eliminate the value of the stress test; if subgroup disparities emerge even in this structured formalization, they warrant closer scrutiny in richer, more realistic committee settings."*

---

> > ### Author Rebuttal · Reviewer_P1Px · 2026-03-31
> >
> > Thanks to the authors for their reply. I'll keep my score.

---

### Official Review · Reviewer_AQpy · 2026-03-09

**Significance:** 4
**Argument Clarity:** 4
**Rating:** 5
**Confidence:** 3

**Questions:**

\

**Alternative Views Section:**

Yes

**Compliance With Llm Reviewing Policy A Conservative:**

Affirmed.

**Discussion Potential:**

3

**Paper Summary:**

The paper argues that without equity and explainability as core design constraints, such systems will exacerbate healthcare disparities. Using empirical evidence from a multi-agent simulation of a liver transplant selection committee, the paper demonstrates that even high-performing agents can systematically disadvantage patients based on sex, ethnicity, and socioeconomic status. The authors further contend that without fairness-aware deployment strategies, such systems cannot be reliably audited or ethically integrated into real-world care. Moreover, they propose a technical roadmap with subgroup-sensitive learning objectives, counterfactual reasoning modules, clinician-in-the-loop governance,
and deployment protocols that address the digital divide.

**Position:**

Yes

**Position In Title:**

Yes

**Related Work:**

3

**Strengths And Weaknesses:**

Strengths:
- The studied topic is relevant.
- The paper provides clear argument for their position.
- The empirical evidence is hintful.

**Support:**

4

---

> ### Author Rebuttal · Authors · 2026-03-31
>
> We thank the reviewer for the positive assessment and for recognizing the relevance of the topic, the clarity of the argument, and the value of the empirical evidence. We are encouraged that the paper’s central contribution came through clearly, and in the camera-ready version we will further sharpen the framing around deployment equity and the role of the roadmap as a forward-looking design agenda.

---

> > ### Author Rebuttal · Reviewer_AQpy · 2026-04-02
> >
> > Thank you for the reply

---

### Official Review · Reviewer_ucfB · 2026-03-16

**Significance:** 2
**Argument Clarity:** 3
**Ethics Flag:** Yes
**Rating:** 3
**Confidence:** 4

**Questions:**

This work probably will not be useful for a fairly long time. In real world,  the problem of "Who Gets an Organ" will not be depended on AI decisions. First, which patient will be benefited most from the available organ is not a scientifically solved problem yet by AI (even the current clinical rules are more or less basic), maybe very far by considering the essnetial complexity. Second, how the AI decision will be judged as "fair/unbiased/interpretable" for organ transplant? This is also very complex, at least much more than the content covered in this paper. So this paper is still quite a distance on solving this problem.

**Alternative Views Section:**

Yes

**Compliance With Llm Reviewing Policy A Conservative:**

Affirmed.

**Discussion Potential:**

3

**Ethical Review Concerns:**

The decision of Who gets an organ should still be remained for human experts to decide, with clear rules.

**Ethics Review Area:**

["Discrimination / Bias / Fairness Concerns"]

**Paper Summary:**

This paper explores potential risks of multi-agent AI systems based on large language models in transplant medicine decision-making, particularly regarding issues of fairness and interpretability. The authors conduct an experiment with a multi-agent system. It simulates a liver transplant selection committee. Results show that even high-performing agents may systematically disadvantage patients based on gender, race, and socioeconomic status. The paper identifies three major failure modes: role-specific bias propagation, opaque decision-making logic, and the digital divide in deployment. It also proposes a technical roadmap, covering data modeling, agent architecture, an explainability stack, and fairness auditing.

**Position:**

Yes

**Position In Title:**

Yes

**Related Work:**

3

**Strengths And Weaknesses:**

Strengths

1. Organ transplantation represents a critical, high-stakes domain. This case study therefore strengthens the paper's urgency and ethical relevance.
2. The paper proposes five core pillars: data modeling, agent architecture, explainability stack, fairness auditing, and deployment architecture. These provide concrete technical guidance for building fair and explainable multi-agent systems, including causal inference modules and counterfactual explanations.
3. The paper clearly shows how agents exploit non-clinical proxy variables, such as insurance type and educational level, to introduce bias. This insight offers valuable lessons for data preprocessing and feature engineering.

Weaknesses

1. The study is primarily simulation-based. Although it utilizes SRTR data, the agents' decision-making processes do not reflect those of real-world physicians. Real clinical settings is needed to validate the effectiveness of the proposed roadmap.
2. The paper notes that equity constraints may impose an "accuracy tax." However, the discussion could go further: how should equity metrics be balanced with medical utility in resource-constrained scenarios, such as organ shortages?
3. The proposed roadmap relies on complex components that may be difficult for resource-constrained hospitals to adopt. While the paper acknowledges the digital divide, it offers limited guidance on lowering deployment barriers for these advanced features.
4. The paper references the EU AI Act and FDA guidelines. Yet it lacks a detailed strategic analysis of how multi-agent systems can navigate specific regulatory approval processes to ensure compliance.

**Support:**

2

---

> ### Author Rebuttal · Authors · 2026-03-31
>
> We thank the reviewer for the thoughtful engagement and for recognizing the clinical urgency of the transplant setting, the value of the roadmap, and the importance of our findings on proxy-variable reliance. We address the main concerns directly.
>
> **1) Empirical grounding, simulation, and intended use.**
> A key clarification is that the patient cohort is not simulated: it is built from real SRTR liver transplant recipient data with real observed covariates and real post-transplant outcomes used as ground truth. The synthetic component is limited to constructing negative/control cases, because SRTR structurally contains waitlisted/transplanted patients rather than declined candidates. To remove ambiguity, we will state this explicitly in the data/methods section: *“Our study uses a hybrid empirical-simulation design: real national registry patients and outcomes, with synthetic negative-case construction only where registry structure makes declined candidates unavailable.”* This matters because the subgroup disparities arise from AI adjudication over real clinical profiles, not from a fully artificial cohort.
>
> We also clarify that, while the multi-agent committee was not designed to reproduce every nuance of physician reasoning and we do not claim prospective clinical validity, its structure was not an abstract design choice. As described in Lines 127–132, the four agent roles were selected to mirror the multidisciplinary composition of real transplant selection committees and the major reasoning domains represented in committee deliberation. The AI-SC is therefore intended as a pre-deployment stress test and risk-discovery framework, surfacing failure modes before patient-facing use rather than replacing human judgment. To make this boundary unmistakable, we will add: *“The AI-SC is presented here as a decision-support stress-test and risk-discovery framework, not as a replacement for human transplant committee judgment.”* Final decisions must remain with human experts, and nothing in our paper argues otherwise.
>
> More broadly, our claim is not that the current system is deployment-ready, but that pre-deployment stress testing can reveal subgroup-specific failure modes before real-world use. If such failures appear even under controlled conditions, that strengthens the case for treating fairness and interpretability as design-time requirements rather than post hoc additions.
>
> **2) Equity-utility tradeoff (“accuracy tax”).**
> We agree this tension deserves sharper framing. Our position is not that fairness must always appear as an optimization penalty trading off against utility. Some protections operate as deployment gates and monitoring requirements rather than model-training constraints. To make this operational, we will add: *“In high-stakes deployment, subgroup calibration checks, disparate-impact monitoring, decision logging, and mandatory clinician override pathways can function as safety gates without requiring every fairness safeguard to be encoded as a direct utility tradeoff during model optimization.”* This makes clear that part of the fairness agenda we propose concerns safe deployment practice, not only optimization-time penalties.
>
> **3) Complexity of the roadmap for under-resourced hospitals.**
> We agree this is the strongest systems-level concern. The current draft acknowledges the digital divide, but it should state more explicitly that the roadmap is tiered rather than uniform. We will add: *“The proposed architecture is a staged menu, not a single deployment requirement: a minimum viable fairness tier, subgroup auditing, disparate-impact monitoring, clinician override, and audit logging, is feasible for lower-resource centers, whereas components such as Bayesian uncertainty heads or structural causal modules represent advanced layers for well-resourced institutions.”*
>
> **4) Regulatory discussion.**
> The references to the EU AI Act and FDA guidance are included as motivating context for why explainability-by-design and auditability matter in this domain; they are not intended to constitute a full regulatory approval roadmap. To avoid over-reading, we will state this more precisely: *“Our discussion of regulatory frameworks is included to motivate design-time requirements for transparency and accountability, not to claim a complete approval strategy for multi-agent systems in transplant medicine.”*
>
> **5) Practical usefulness and timeline.**
>
> Our paper does not advocate autonomous AI decision-making in organ allocation. Rather, our argument is prospective and precautionary: if multi-agent AI is being actively developed to assist high-stakes clinical committee processes, then fairness, interpretability, and deployment equity need to be established as design-time constraints before such systems mature into clinical workflow candidates. The contribution is therefore not a claim of immediate deployment readiness, but a claim that pre-deployment failure analysis is necessary early, not late.

---

> > ### Author Rebuttal · Reviewer_ucfB · 2026-04-06
> >
> > This is an exciting topic, but I still have major technical concerns on the actual utility and usefulness of this work, especially by taking into account the realworld clinical complexity. I still think the current version of this work is misleading.

---

### Official Review · Reviewer_jbT8 · 2026-03-24

**Significance:** 2
**Argument Clarity:** 3
**Rating:** 4
**Confidence:** 3

**Questions:**

None

**Alternative Views Section:**

Yes

**Compliance With Llm Reviewing Policy A Conservative:**

Affirmed.

**Discussion Potential:**

2

**Final Justification:**

The rebuttal cleared my concern.

**Paper Summary:**

The paper states that when using agentic ai for sensitive medical task like orgam transplant, it risks amplifying bias and causing unfairness. The author then proposes a framework for an explainable deployment of agentic AI in such tasks.

**Position:**

Yes

**Position In Title:**

Yes

**Related Work:**

3

**Strengths And Weaknesses:**

Strength:
1. Clear identification of failure mode.
2. Clear articulation of the proposed framework.

Weakness:
I think the main weakness is the authors identified 3 failure modes but in the proposed framework, the third one on the digital divide is getting ignored. Another small weakness is the title is very narrow scoped into organ transplant but it seems the position can be applied more broadly.

**Support:**

3

---

> ### Author Rebuttal · Authors · 2026-03-31
>
> We thank the reviewer for recognizing the clarity of our failure mode identification and for raising two constructive concerns, which we address directly below.
>
> **1) On the digital divide and deployment roadmap linkage.**
>
> We agree that the connection between Failure Mode 3, the digital divide
> in Section 3.3, and the deployment roadmap in Section 4.5 is already reflected in the current draft, but remains too implicit. In revision,
> we will make two concrete edits to make that linkage unmistakable.
>
> (1) At the start of Section 4.5, we will add an explicit opening
> sentence stating that this subsection is the roadmap response to the
> deployment inequities identified in Section 3.3. We will add: *"This
> subsection addresses the deployment inequities identified in Section
> 3.3, including uneven infrastructure, proprietary API-cost barriers,
> and limited local technical support in under-resourced settings."*
>
> (2) We will also add a sentence clarifying that the deployment roadmap
> is intended as a tiered framework rather than a uniform requirement.
> We will add: *"A minimum viable fairness tier, centered on subgroup
> auditing, disparate-impact monitoring, and clinician override
> mechanisms, is feasible for lower-resource centers; more
> resource-intensive components are intended for well-resourced
> institutions."*
>
> Together, these edits make clear that the digital divide is not outside
> the framework, but one of the central deployment constraints the
> framework is designed to address. We also acknowledge the tension that
> some advanced roadmap components require substantial compute, which is
> precisely why the tiered distinction matters.
>
> **2) On the title scope.**
>
> We framed the paper around transplant selection because it is a
> particularly high-stakes, rationed, multidisciplinary, and irreversible
> decision context, making it a useful setting in which failure modes
> become especially visible. Our intention was to ground the argument in
> a concrete domain where the stakes are clear, not to suggest that the
> contribution is limited to transplant medicine alone. To make that
> broader relevance clearer, we will add: *"The design principles and
> failure modes discussed here extend to settings such as ICU triage,
> oncology treatment selection, and psychiatric decision-making."*
>
> We hope these targeted changes clarify that the reviewer's concerns
> are addressable through framing and positioning rather than requiring
> a change to the core argument.

---

> > ### Author Rebuttal · Reviewer_jbT8 · 2026-04-05
> >
> > I will change my score accordinly after the rebuttal.

---

### Decision · Program_Chairs · 2026-04-30

**Decision:**

Accept (regular)

**Comment:**

This position is both timely and relvant, given the rapid adoption of multi-agent settings and their potential use in medical settings. There is consensus among reviewers that while these concerns are broadly applicable the use of medicine as an exemplar setting is compelling. Despite this interest, there are several concerns raised during review, many of which anchor to the medical setting as described. In particular, since the position is motivated by a simulation-based study there are concerns that the representational decision-making process doesn't not reflect those of real-world transplant physicians. Further, even if it did reflect stated decision-making processes, it would fail to capture the numerous exogenous factors present in current organ transplant settings. A second concern regards the framing of the position which is scoped tightly to transplant medicine but reads and being more broadly applicable. This may limit engagement from the clininical subcommunity, which may perceive the position is too broad for the stated scope, and the general community, which may have preferred a broader scope us more narrow case study framing of the transplant medicine. Taken together, this positions reception may be limited.